# Uptake of Influenza Vaccine and Factors Associated with Influenza Vaccination among Healthcare Workers in Tertiary Care Hospitals in Bangladesh: A Multicenter Cross-Sectional Study

**DOI:** 10.3390/vaccines11020360

**Published:** 2023-02-05

**Authors:** Md. Mahabub Ul Anwar, Shariful Amin Sumon, Tahrima Mohsin Mohona, Aninda Rahman, Syed Abul Hassan Md Abdullah, Md. Saiful Islam, Md. Golam Dostogir Harun

**Affiliations:** 1Office of Health Affairs, West Virginia University, Morgantown, WV 26506, USA; 2Infectious Diseases Division, International Centre for Diarrhoeal Disease Research, Bangladesh (ICDDR, B), Dhaka 1212, Bangladesh; 3Communicable Disease Control (CDC), Directorate General of Health Services, Government of Bangladesh, Dhaka 1212, Bangladesh; 4SafetyNet Bangladesh, Dhaka 1212, Bangladesh; 5National Centre for Epidemiology and Population Health, Australian National University, Canberra 2601, Australia

**Keywords:** influenza, hospitals, vaccination, healthcare workers, Bangladesh

## Abstract

Influenza, highly contagious in hospital settings, imposes a substantial disease burden globally, and influenza vaccination is critical for healthcare workers (HCWs) to prevent this illness. This study assessed influenza vaccine uptake, including its associated factors among HCWs of tertiary care hospitals in Bangladesh. Between September and December 2020, this multicenter study included 2046 HCWs from 11 hospitals. Face-to-face interviews were conducted using a semi-structured questionnaire to collect data from physicians, nurses, and cleaning and administrative staff for the survey. Only 13.8% (283/2046) of HCWs received the influenza vaccine, of which the majority (76.7%, 217/283) received it for free from the hospital. Nurses had the highest (20.0%, 187/934) influenza vaccine coverage, followed by physicians at 13.5% (71/526), whereas cleaning staff had the lowest at 6.0% (19/318). Among unvaccinated HCWs, the desire to get vaccinated was high (86.2%), with half of the respondents even being willing to pay for it. The HCWs who were aware of the influenza vaccine were over five times more likely to get the vaccine (OR 5.63; 95% CI: 1.04, 1.88) compared to those who were not. HCWs in Bangladesh were vaccinated against influenza at a very low rate. Free and mandatory influenza vaccination programs should be initiated to optimize vaccine coverage among HCWs.

## 1. Introduction

Influenza imposes a substantial disease burden globally, and healthcare workers (HCWs) are the people most vulnerable to influenza transmission due to their work environment and job activities [1]. Simultaneously, HCWs can increase the risk of transmitting the infection to vulnerable patients, visitors, coworkers, and their family members [2,3,4]. Vaccination is one of the most effective methods of avoiding the spread of various viral and bacterial infections and has significantly reduced morbidity and death in recent decades [5]. HCWs who are vaccinated act as a barrier against the infection spreading as well and at the same time ensure essential healthcare service delivery even during emergencies and outbreaks [6,7]. However, the uptake of vaccines among HCWs remains low due to incomplete knowledge and a lack of evidence-based recommendations [8,9]. Findings from different studies have revealed that though healthcare workers know about the perks of becoming immunized, most of them have some trust concerns regarding health authorities [10]. The proportion of these HCWs remains unknown to an extent. Further investigations need to explore the reason for the unwillingness among HCWs when vaccination comes into question and take up necessary measures to resolve the situation [8].

Nosocomial infections caused by influenza are also a critical problem, particularly among immune-compromised patients (hospitalized, elderly, or suffering from chronic degenerative disorders) that add to morbidity and mortality [11]. Each influenza season, about 20% of HCWs are exposed to such infections, which not only give rise to epidemics of hospital-acquired infections (HAIs) but also disrupt the continuation of healthcare service provision [12]. Such infection epidemics could precede the spread of influenza among the general population as well [13]. During the 2009 flu pandemic, HCWs faced a particular risk of influenza A infection, with a pooled prevalence of 6.3% [14]. The World Health Organization (WHO) advisory committee on immunization recommended that HCWs be vaccinated annually against influenza [15]. Still, in European countries, influenza vaccination uptake remained comparatively very low; not even a single member state in Europe reached the 75% coverage limit during 2014–15 [16]. Even countries implementing massive immunization campaigns have as low as 42% influenza vaccination coverage [17]. Mainly, the attitudes and beliefs of HCWs towards vaccination for influenza have been found in many studies to be critical determinants in the decision-making process, whereas risk perceptions were termed to be vaccine propensity predictors [18,19,20]. Despite a large amount of published research on influenza infection, there is a lack of credible information on influenza infection among HCWs in Bangladesh, including infection epidemiology [21]. In addition, vaccination coverage against this preventable disease among HCWs in Bangladesh is also unsatisfactory. Therefore, this study aims to determine the prevalence of influenza vaccine uptake, intention to receive the vaccine, and factors that affect the uptake among HCWs of selected tertiary care hospitals in Bangladesh. The findings would help to point out gaps in the healthcare system and to inform policymakers and programmers of the ongoing situation so that they can take prompt and necessary actions to diminish the burden of this vaccine-preventable disease. 

## 2. Materials and Methods

### 2.1. Study Design and Settings 

From September to December 2020, we conducted this multicenter cross-sectional survey at 11 tertiary care hospitals across the country. We purposively selected all study sites based on the Ministry of Health’s recommendation; nine of the hospitals were government-operated, and two were private healthcare facilities. The average bed capacity and annual patient turnover ranged from 450–2600 and 15,000–85,000, respectively. The selected healthcare facilities have specialized departments to serve as referral hospitals and represent one-quarter of all tertiary hospitals in Bangladesh.

### 2.2. Participants

Participants in the study included physicians, nurses, cleaning staff, and administrative staff who worked in both clinical and non-clinical roles and provided direct or indirect patient care in selected hospitals. Before conducting the survey, written permission was collected from the Directorate General of Health Services (DGHS) of Bangladesh, and the official approval letter was distributed to each survey hospital. We approached the HCWs for an interview after obtaining permission from the respective hospital administrations. Face-to-face interviews with participants in the local (Bengali) language were used to collect data.

### 2.3. Sample Size and Sampling Procedures

We enrolled participants proportionately from all respondent groups and hospitals. We collected respondent-specific total employee numbers from each hospital and randomly interviewed 25% of each respondent category. We enrolled a total of 2046 HCWs for this study, comprising 526 physicians, 934 nurses, 268 cleaning workers, and 318 administrative staff. We considered both permanent and temporary staff who were available at the time of data collection and willing to participate.

### 2.4. Data Collection

We used a semi-structured questionnaire to collect data on influenza vaccination status. In addition to basic demographic information, we collected data on influenza vaccine uptake, awareness about the influenza vaccine and availability, their intention and willingness to pay if unimmunized, and their opinions on vaccination uptake in terms of infection prevention and control measures. Moreover, we inquired about a history of influenza-like illness (ILI) and the number of times they suffered from ILI in the past 12 months. We considered respondents’ recall of influenza vaccination uptake, history, and incidences of ILI during data collection. Prior to conducting the interviews, we informed participants about the study’s purpose, voluntary participation, and their rights to participate and assured them of the strict confidentiality, privacy, and anonymity of the collected information. 

### 2.5. Statistical Analyses

We performed both descriptive and multivariate analyses to present the findings. We described categorical and numerical variables with frequency, percentage, mean, and standard deviation (SD). We used multivariate logistic regression models to determine the association between influenza vaccination uptake and type of HCW, age group (18–30, 31–40, 41–50, and ≥51 years), familiarity with influenza vaccine, and vaccine recommendation. Multicollinearity between independent variables (HCW type, sex, age, educational qualification, working experience, ownership type of hospital, etc.) was checked, and variables with a *p*-value of ≥0.25 in the univariate model were considered for the final model. We presented the multivariate results as an adjusted odds ratio (AOR) with a 95% confidence interval (CI) and considered *p* < 0.05 to be statistically significant. We used STATA (version 13.1) for all analyses. 

### 2.6. Quality Control

To ensure the quality of the data, experienced data collectors were recruited and trained in hands-on data collection. Data collection was also supervised by trained professionals. Research experts oversaw the study design, data collection guidelines and procedures, quality control planning, and data management and analysis. The respondents were given enough time during the interview to recall the information on vaccination and ILI.

## 3. Results

### 3.1. Basic Characteristics of Participating HCWs

A total of 2130 healthcare workers were approached, of whom 2046 were interviewed with a response rate of 96.0%. The basic characteristics of the study participants are described in Table 1. The majority of the respondents (45.6%) were nurses, followed by physicians (25.7%), administrative staff (15.5%), and cleaning staff (13.1%). The median age of vaccinated respondents was 29 (IQR: 25–36), with a median of 5 years (IQR: 2–11) of working experience. The overall influenza vaccination uptake was 13.8% (283/2046, 95% CI: 12.3, 15.4) among HCWs. Nurses had the highest coverage (20.0%, 187/934), followed by physicians with 13.5% (71/526). However, only 2.2% (96/268) of cleaning staff had received the influenza vaccine. Influenza vaccination coverage among HCWs in public hospitals was higher (14.4%) than in private hospitals (10.0%).

### 3.2. Acquaintance and Preference Regarding Influenza Vaccination

Table 2 depicts the HCWs’ acquaintance and preferences toward influenza vaccination. Overall, two-thirds (65.9%, 95% CI: 63.8, 67.9) of the HCWs were knowledgeable about influenza vaccination, but only one-third of the cleaning and administrative staff were aware of the influenza vaccine. Among all recipients of the influenza vaccine, 76.7% (217/283, 95% CI: 74.8, 78.5) had received it for free from their respective hospitals, and the remaining recipients purchased it from a local pharmacy. Only half of the vaccinated physicians (50.7%, 36/71) benefited from a free influenza vaccine from the government. Around 88.8% (1566/1763, 95% CI: 87.4, 90.1) of non-recipients desired to take the influenza vaccine in the future, and half of them (50.0%, 783/1566) were willing to pay to get vaccinated. In the last year, one-third (29.7%) of HCWs reported having experienced any influenza-like symptoms, with a mean of 1.9 times (SD 1.0). Among them, half of the support staff and administrative staff had a history of influenza-like symptoms occurring at least twice in the past year. Regarding HCWs’ susceptibility to influenza infection, 67.3% of participants knew it was infectious, but only 34.8% (95% CI: 32.1, 36.3) recommended the influenza vaccine. However, the vast majority of participants (90.2%) consider the influenza vaccine crucial for protecting HCWs against influenza. Additionally, 90.5% (95% CI: 89.2, 91.8) of respondents opined that influenza vaccination should be mandatory for healthcare providers in our country.

### 3.3. Associated Factors of Influenza Vaccination among HCWs

Table 3 displays the factors associated with influenza vaccination. The HCWs who were aware of influenza immunization had a greater likelihood (AOR: 5.64) of receiving the influenza vaccine (95% CI: 3.54, 8.97, *p*-value: <0.001) than those who were not. Physicians, administrative staff, and cleaning staff had significantly lower uptake of influenza vaccine compared to nurses. Among age groups, those aged 31–40 years and 41–50 years were 50% less likely to receive the influenza vaccine (AOR: 0.52, 95% CI: 0.37, 0.71, and AOR: 0.53, 95% CI: 0.34, 0.82 respectively) than those aged 18–30 years. Regarding suggestions for a vaccine in the context of Bangladesh, the HCWs who recommended the influenza vaccine were 1.47 times more likely to obtain the influenza vaccine (95% CI: 1.12, 1.92). In contrast, those who considered HCWs susceptible to influenza infection had lower odds (AOR: 0.58, 95% CI: 0.43, 0.76) of getting the influenza vaccine than the reference category (those who were not considered HCWs susceptible to influenza infection). Participants’ sex did not affect influenza vaccination coverage.

## 4. Discussion

This multicenter study identified and filled the gap in the current influenza vaccination uptake literature for HCWs in Bangladesh. The results showed that influenza vaccination uptake among HCWs was remarkably low, particularly among cleaning and administrative staff. However, most HCWs are eager to get the flu vaccine, and they are even willing to pay for it. The majority of the HCWs opined that vaccination is very important and recommended mandatory and free influenza vaccination for all health service providers. 

Our study revealed a low overall uptake (13.8%) of influenza vaccination among HCWs in tertiary hospitals in Bangladesh. This finding was consistent with prior studies in similar LMIC settings that found poor influenza and other vaccine uptake across hospital staff [17,22]. Even so, this uptake rate was lower than in other LMICs, such as Egypt (30.7%) and Lebanon (40.4%) [23,24]. According to this study, cleaning and administrative staff had poorer vaccination coverage compared to nurses and physicians. This lower level of employees, such as cleaning staff, is often neglected regarding preventive approaches such as vaccination. Recent research in Bangladesh found that cleaning personnel were generally overlooked when it came to acquiring vaccinations [25]. This implies that a change in mindset is required to incorporate cleaning staff into the mainstream of all preventive approaches. Most (76.7%, 217/283) vaccine recipients received their influenza vaccinations free from government sources. The government of Bangladesh provided free vaccines to some HCWs, particularly physicians and nurses who were deployed on special duties on pilgrimage (hajj) in Saudi Arabia. This finding was in line with previous studies which reported that free vaccination programs increase the vaccination coverage among HCWs in both developed and developing countries [26,27,28]. Two studies conducted in the United States reported increases in uptake of 24% and 41% when free vaccines were provided [29,30]. The Centers for Diseases Control and Prevention (CDC) recommends flu vaccination be mandatory for HCWs to prevent the spread of the infection between patients and workers, and vice versa [31]. Our study highlights the importance of providing free vaccines, as further initiative needs to be taken by the government to implement on-site, free-of-cost mass vaccination campaigns. Physicians constituted the highest percentage of HCWs who had paid for the vaccine. This finding is consistent with a study which found that physicians were much more likely to get vaccinated to protect both themselves and their patients and had a higher buying capacity compared to other HCWs [32]. This finding might also be interlinked with physicians’ greater knowledge and awareness of vaccines. This study found that knowledge on the benefits of the influenza vaccine and perceptions of influenza severity was significantly associated with vaccination coverage among HCWs. This finding is consistent with other studies which reported that higher knowledge and perceived severity of influenza enhance vaccine uptake among HCWs [33,34,35,36].

The study found that the desire to take the vaccine was high across all the unvaccinated HCW groups, with half of the respondents even being willing to pay for the flu vaccine. This finding is in stark contrast to a Tunisian study where more than 50% of participating HCWs were reluctant to take a vaccine, even if it was offered for free [26]. In terms of awareness about the influenza vaccine, administrative and cleaning staff were found to be the least aware. Our analyses revealed a strong association between HCWs’ awareness of flu vaccination and vaccine uptake. This suggests a need to understand obstacles that may be specific to distinct subgroups among HCWs [37,38]. The result highlights the importance of improving focused communication and information dissemination to the specific groups of HCWs with poor vaccination uptake rates. Vaccination campaigns should not be limited to hospital settings for HCWs but should also include other risky populations such as pregnant women and the aged population. Regarding influenza vaccine uptake, previous studies in other resource-limited settings showed sub-standard coverage where lack of awareness and accessibility, misconceptions, and high cost were identified as key barriers [22,39,40]. In addition, as the influenza vaccine needs to be administered annually and costs range from USD 12–30, it might not be financially feasible for cleaning staff with a lower salary structure to get vaccinated every year [41]. These findings further imply the importance of raising vaccination awareness among HCWs and organizing free vaccination programs.

Our study showed that almost one-third of HCWs reported influenza-like illness during the previous year. These HCWs might further transmit the virus to patients and visitors in hospital settings. This can create greater difficulty in a densely populated country like Bangladesh [42]. Prior evidence has shown that flu vaccines decrease the risk of influenza-like illnesses among HCWs and inpatients in hospital settings [43]. Therefore, the vaccination of HCWs is critical in reducing influenza-like illness in HCWs, patients, and visitors [3,44]. Influenza vaccination coverage among HCWs in public hospitals was higher than in private hospitals (14.4% vs. 10.0%). This finding was consistent with a study conducted in Hong Kong that found poor vaccine coverage in private health facilities [45]. This finding implies that along with the government, the onus for improving vaccination coverage in HCWs is also on private facilities in Bangladesh, as they need to ensure improved flu vaccination coverage [46]. This multicenter study was conducted in 11 tertiary care hospitals across the country, with a large sample size (2046 HCWs). Our study included both public and private hospitals, as well as all levels of HCWs, i.e., physicians, nurses, cleaning staff, and administrative staff. These characteristics allowed for a more representative sample, which will help generate evidence that policymakers can use to formulate a practical, comprehensive strategy.

Our study had some limitations as well. First, self-reported data on vaccination history, without verifying a vaccine card or proof of vaccination place, and incidences of ILI were used, which might have resulted in recall bias. Second, the hospitals were purposively selected, which may have been subject to selection bias. Third, the participants in the present study were from 11 tertiary care hospitals in Bangladesh. Thus, the conclusions for flu vaccination status among HCWs may not be generalized to the entire country. In addition, our study did not investigate the causal relationship due to cross-sectional assessment and vaccination uptake barriers among HCWs.

## 5. Conclusions

Influenza vaccination uptake among HCWs in Bangladesh remains low. Considering the importance of protection against influenza, an upsurge in the influenza vaccine uptake needs to be immediately pursued. Our findings revealed a sub-standard uptake of vaccines across all HCWs, especially among the cleaning and administrative staff. Appropriate evidence-based, tailored intervention needs to be part of future campaigns to improve knowledge and awareness among HCWs concerning the importance of the vaccine. Special attention needs to be given to cleaning and administrative staff, and a shift in mindset is necessary to ensure substantial improvement. In our study, most unvaccinated HCWs were interested in receiving the influenza vaccine and opined in favor of free vaccination. The government of Bangladesh needs to implement mandatory and free-of-cost vaccination programs, which could be established as part of employee health and safety policies. Despite the strong desire for the influenza vaccine among workers, the vaccine uptake was low. Future studies aimed at exploring why there was a gap between vaccine demand and uptake could be conducted to feed future vaccination campaigns. 

## Figures and Tables

**Table 1 vaccines-11-00360-t001:** Basic characteristics of study participants.

Variables	Total Healthcare Workers (*N* = 2046)	Total Influenza Vaccine Uptake (*N* = 283)	Proportion Vaccinated (%)
		*n*/*N* (%)	
Overall	2046	283	13.8%
Type of HCWs			
Physicians	526 (25.7)	71 (25.1)	13.5%
Nurses	934 (45.6)	187 (66.1)	20.0%
Support Staff	268 (13.1)	6 (2.1)	2.2%
Administrative staff	318 (15.5)	19 (6.7)	6.0%
Gender of HCWs			
Female	1274 (62.3)	216 (76.3)	17.0%
Male	772 (37.7)	67 (23.7)	8.7%
Age in Years			
Median (IQR *)	32 (27–40)	29 (25–36)	
<30	923 (45.1)	173 (61.1)	18.7%
31–40	656 (32.1)	64 (22.6)	9.8%
41–50	314 (15.4)	28 (9.9)	8.9%
≥51	153 (7.5)	18 (6.4)	11.8%
Education of HCWs			
Master’s degree/above	548 (26.8)	75 (26.5)	13.7%
Bachelor’s degree/Diploma in nursing	935 (45.7)	187 (66.1)	20.0%
Higher secondary	120 (5.9)	11 (3.9)	9.2%
≥Secondary l	443 (21.6)	10 (3.5)	2.3%
Working experience of HCWs			
Median (IQR *)	7 (3–16)	5 (2–11)	
≤2 years	415 (20.3)	75 (26.5)	18.1%
3–5 years	447 (21.9)	74 (26.1)	16.5%
6–10 years	464 (22.7)	64 (22.6)	13.8%
11–15 years	173 (8.5)	15 (5.3)	8.7%
≥16 years	547 (26.7)	55 (19.4)	10.1%
Type of health facility			
Public hospital	1795 (87.7)	258 (91.2)	14.4%
Private hospital	251 (12.3)	25 (8.8)	10.0%

* IQR: Interquartile range.

**Table 2 vaccines-11-00360-t002:** Perception and practices regarding influenza vaccination among healthcare workers.

Variables	Physician (*N* = 526)	Nurse (*N* = 934)	Cleaning Staff (*N* = 268)	Administrative Staff (*N* = 318)	Total (*N* = 2046)
			*n/N* (%)		
Know about influenza vaccination	444 (84.4)	708 (75.8)	80 (29.8)	116 (36.5)	1348 (65.9)
Suffering from any influenza-like illness (ILI) within the last year	199 (37.8)	259 (27.7)	86 (32.1)	64 (20.1)	608 (29.7)
Number of times suffered ILI in the past year					
Mean ± SD	1.9 ± 1.1	1.9 ± 1.1	1.8 ± 0.8	1.7 ± 0.7	1.9 ± 1.0
1 Time	88 (44.2)	109 (42.1)	31 (36.1)	21 (32.8)	249 (12.2)
2 Times	67 (33.7)	93 (35.9)	43 (50.0)	33 (51.6)	236 (11.5)
>3 Times	44 (22.1)	57 (22.0)	12 (13.9)	10 (15.5)	123 (6.0)
Never experienced	327 (62.2)	675 (72.3)	182 (67.9)	254 (79.9)	1438 (70.3)
Place of getting the vaccine (among vaccinated HCWs)	*N* = 71	*N* = 187	*N* = 6	*N* = 19	*N* = 283
Free from hospital	36 (50.7)	160 (85.6)	5 (83.3)	16 (84.2)	217 (76.7)
Purchased own from outside	35 (49.3)	27 (14.4)	1 (16.7)	3 (15.8)	66 (23.3)
The desire for taking influenza vaccine upon availability (among unvaccinated HCWs)	*N* = 455	*N* = 747	*N* = 262	*N* = 299	*N* = 1763
	371 (81.5)	682 (91.3)	242 (92.4)	271 (90.6)	1566 (88.8)
Willingness to pay for influenza vaccine (among HCWs who desired to receive vaccine)	*N* = 371	*N* = 682	*N* = 242	*N* = 299	*N* = 1566
	212 (57.1)	411 (60.3)	63 (26.0)	97 (32.4)	783 (50.0)
Healthcare workers (HCWs) susceptible to influenza infections	362 (68.8)	658 (70.5)	165 (61.6)	192 (60.4)	1377 (67.3)
Recommended influenza vaccine for HCWs	225 (42.8)	365 (39.1)	43 (16.0)	66 (20.7)	699 (34.2)
Perceived importance of influenza vaccine	477 (90.7)	892 (95.5)	222 (82.8)	261 (82.1)	1852 (90.5)
Should be compulsory to get the influenza vaccine for all HCWs in our country	464 (88.2)	898 (96.1)	221 (82.5)	263 (82.7)	1846 (90.2)

**Table 3 vaccines-11-00360-t003:** Association between influenza vaccination uptake and multiple variables among vaccinated HCWs.

Variables	Univariate Model		Multivariate Model	
	Crude Odds Ratio (95% CI)	*p*-Value	Adjusted Odds Ratio (95% CI)	*p*-Value
Type of HCWs				
Nurses	Reference		Reference	
Physicians	0.62 (0.46, 0.84)	0.002	0.53 (0.37, 0.76)	0.001
Support Staff	0.09 (0.04, 0.21)	<0.001	0.18 (0.07, 0.43)	<0.001
Administrative staff	0.25 (0.15, 0.41)	<0.001	0.46 (0.26, 0.83)	0.009
Gender of HCWs				
Male	Reference		Reference	
Female	2.15 (1.61, 2.87)	<0.001	1.08 (0.74, 1.59)	0.683
Age in years				
18–30	Reference		Reference	
31–40	0.47 (0.35, 0.64)	<0.001	0.52 (0.37, 0.71)	<0.001
41–50	0.42 (0.28, 0.65)	<0.001	0.53 (0.34, 0.82)	0.005
≥51	0.58 (0.34, 0.97)	0.038	0.86 (0.49, 1.51)	0.599
Familiar with influenza vaccination				
No	Reference		Reference	
Yes	7.38 (4.72, 11.52)	<0.001	5.64 (3.54, 8.97)	<0.001
Suggestion of influenza vaccine for HCWs				
No	Reference		Reference	
Yes	1.81 (1.40, 2.33)	<0.001	1.47 (1.12, 1.92)	0.005
HCWs are susceptible to influenza				
No	Reference		Reference	
Yes	0.81 (0.62, 1.05)	0.118	0.58 (0.43, 0.76)	<0.001

## Data Availability

The authors of this survey are responsible for the data described in this manuscript. The datasets that were generated and analyzed are available upon request from the corresponding author.

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
