# Peer review of "Uptake of Influenza Vaccine and Factors Associated with Influenza Vaccination among Healthcare Workers in Tertiary Care Hospitals in Bangladesh: A Multicenter Cross-Sectional Study"

_vaccines, 2023, doi:10.3390/vaccines11020360_

Round 1

Reviewer 1 Report

Title:

“Uptake of influenza vaccine and factors associated with influenza vaccination among healthcare workers in tertiary care hospitals in Bangladesh: A multicenter cross-sectional study”

Revision

The manuscript is interesting, and the intent of this revision is to improve this manuscript. Some suggestions may be indicated.

1. In this study, to demonstrate that participants are a sample representative of the healthcare workers of tertiary care hospitals in Bangladesh is important. In order to indicate this aspect, a detail explanation of the sampling design, power, α, confidence level, sample size, etc. could be mentioned in the manuscript. For example, see (1)

2. The participation rate is not described, and this is needed for an evaluation of the study. On the other hand, a comparison between participants and non-participants could understand better the study

3. Considering that the design of the study, confidence limits of several outcomes such as the prevalence influenza vaccination may be reported.

4. The potential confounding factors analysed could be indicated in the material and methods.

5. Sex is not included in the logistic regression model (Table 3). However, this variable may be not a significative but may be a confounding factor.

6. In Table 3, all variables were analysed together. This may be considered the nominated “table 2 fallacy” (2). Independent variables (healthcare workers type, age,…..) could have some different confounding factors, and Table 3 may be misunderstood.

7. There are some limitations of this study which are not mentioned in the manuscript, including the logistic models. The construction of multivariable models could be ameliorated with a Direct Acyclic Graphic (DAGs) approach, which permits a better approach to adjust for confounders factors (3, 4). In addition, an objective measure of influenza vaccination is not presented such as a vaccination register.  

8.   References of the manuscript should follow the journal’s reference guidelines.

References

1. Riccio, M.; Marte, M.; Imeshtari, V.; Vezza, F.; Barletta, V.I.; Shaholli, D.; Colaprico, C.; Di Chiara, M.; Caresta, E.; Terrin, G.; Papoff, P.; La Torre, G. Analysis of Knowledge, Attitudes and Behaviours of Health Care Workers towards Vaccine-Preventable Diseases and Recommended Vaccinations: An Observational Study in a Teaching Hospital. Vaccines 2023, 11, 196. https://doi.org/10.3390/vaccines11010196

2. Westreich D1, Greenland S. The table 2 fallacy: presenting and interpreting confounder and modifier coefficients. Am J Epidemiol. 2013;177:292-298.

3. Greenland S, Pearl J, Robins JM. Causal diagrams for epidemiologic research. Epidemiology. 1999;10:37–48. pmid:9888278

4. Textor J, van der Zander B, Gilthorpe MS, Liskiewicz M, Ellison GT. Robust causal inference using directed acyclic graphs: the R package ’dagitty’. Int J Epidemiol. 2016;45:1887–1894. pmid:28089956

Author Response

Dear Reviewer;

Thank you very much for your kind review of our manuscript titled "(vaccines-2183666) "Uptake of influenza vaccine and factors associated with influenza vaccination among healthcare workers in tertiary care hospitals in Bangladesh: A multicenter cross-sectional study." We have addressed your comments and resubmitted the revised version of the manuscript. We (Author & Co. Authors) tried our best to concentrate on the comments/ suggestions. We appreciate your thorough reviews and very useful comments. We gratefully acknowledge the recommendations, that was very helpful in enriching our manuscript.

Please find the point-by-point comments and address below.  

  1. In this study, to demonstrate that participants are a sample representative of the healthcare workers of tertiary care hospitals in Bangladesh is important. In order to indicate this aspect, a detail explanation of the sampling design, power, α, confidence level, sample size, etc. could be mentioned in the manuscript. For example, see (1)

Response: Thank you for your suggestion. We interviewed approximately 25% of healthcare workers of each category from every hospital to ensure proper representatives. We have added this in the participant's section of the manuscript, on page 2, lines 134-135.

  1. The participation rate is not described, and this is needed for an evaluation of the study. On the other hand, a comparison between participants and non-participants could understand better the study

Response: Thank you for pointing out this issue. We have included the response rate on page 3, lines 130-132.

  1. Considering that the design of the study, confidence limits of several outcomes such as the prevalence influenza vaccination may be reported.

Response: Thank you. We have included the CI in the line 137, page-3

  1. The potential confounding factors analysed could be indicated in the material and methods.

Response: Thank you. We have added the list of variables in lines 116-117. We cannot tell the confounding in the method section until we analyze it..

  1. Sex is not included in the logistic regression model (Table 3). However, this variable may be not a significative but may be a confounding factor.

Response: Thank you for this analysis suggestion. We have incorporated the sex in the model and revised the analysis accordingly in line 181 and table 3, page 6.

  1. In Table 3, all variables were analysed together. This may be considered the nominated “table 2 fallacy” (2). Independent variables (healthcare workers type, age,…..) could have some different confounding factors, and Table 3 may be misunderstood.

Response: Thank you for your valuable recommendation. Due to the limited number of significant variables, we prefer the model in a combined table.

  1. There are some limitations of this study which are not mentioned in the manuscript, including the logistic models. The construction of multivariable models could be ameliorated with a Direct Acyclic Graphic (DAGs) approach, which permits a better approach to adjust for confounders factors (3, 4). In addition, an objective measure of influenza vaccination is not presented such as a vaccination register.

Response: Thank you so much for this suggestion. We have preferred multivariate logistic regression models after checking the multicollinearity (mentioned on page lines 113-118). We have included the point in the limitation para, page 8, line 257.

  1. References of the manuscript should follow the journal’s reference guidelines.

Response: We have revised the reference list according to the journal guideline.

References

  1. Riccio, M.; Marte, M.; Imeshtari, V.; Vezza, F.; Barletta, V.I.; Shaholli, D.; Colaprico, C.; Di Chiara, M.; Caresta, E.; Terrin, G.; Papoff, P.; La Torre, G. Analysis of Knowledge, Attitudes and Behaviours of Health Care Workers towards Vaccine-Preventable Diseases and Recommended Vaccinations: An Observational Study in a Teaching Hospital. Vaccines 2023, 11, 196. https://doi.org/10.3390/vaccines11010196
  2. Westreich D1, Greenland S. The table 2 fallacy: presenting and interpreting confounder and modifier coefficients. Am J Epidemiol. 2013;177:292-298.
  3. Greenland S, Pearl J, Robins JM. Causal diagrams for epidemiologic research. Epidemiology. 1999;10:37–48. pmid:9888278
  4. Textor J, van der Zander B, Gilthorpe MS, Liskiewicz M, Ellison GT. Robust causal inference using directed acyclic graphs: the R package ’dagitty’. Int J Epidemiol. 2016;45:1887–1894. pmid:28089956

Response: Thank you for sharing the reference list, we have gone through the article for better understanding and as references. 

Submission Date

26 January 2023

Reviewer 2 Report

  1. Material and Methods, section 2.1: Please provide the total number of HCPs in the participating (11) hospitals.
  2. First line of Results: please provide the response rate among HCWs.
  3. Methods: Please provide data about ethical approval of the study ( Number of letter, date issued).
  4. Figure 1: There is no need to have such a Figure, data are already presented in Table 1.
  5. Please provide a separate Limitations section, and expand by adding some more information
  6. Please add the following WHO study in your reference list:     

Maltezou et al. Influenza vaccination policies for health workers in low-income and middle-income countries: A cross-sectional survey, January-March 2020. Vaccine. 2020 Nov 3;38(47):7433-7439.

Author Response

Dear Reviewer;

Thank you very much for your kind review of our manuscript titled "(vaccines-2183666) "Uptake of influenza vaccine and factors associated with influenza vaccination among healthcare workers in tertiary care hospitals in Bangladesh: A multicenter cross-sectional study." We have addressed your comments and resubmitted the revised version of the manuscript. We (Author & Co. Authors) tried our best to concentrate on the comments/ suggestions. We appreciate your thorough reviews and very useful comments. We gratefully acknowledge the recommendations, that was very helpful in enriching our manuscript.

Please find the point by point comments and address below.

Material and Methods, section 2.1: Please provide the total number of HCPs in the participating (11) hospitals.

Response: Thank you for your kind suggestion. We have incorporated this information on page 3, lines 130-132.

First line of Results: please provide the response rate among HCWs.

Response: We have included the response rate in page 3, line-131

  1. Methods: Please provide data about ethical approval of the study (Number of letter, date issued).

Response: Thank you for this opinion. We have clearly mentioned Institutional Review Board and Informed Consent Statement on page 8, lines 283-289.

Figure 1: There is no need to have such a Figure, data are already presented in Table 1.

Response: Thank you, we have excluded the figure from the manuscript.

Please provide a separate Limitations section, and expand by adding some more information

Response: Thank you for the suggestion. We have separated the limitation paragraph and added additional limitations on page 8, line 257.

  1. Please add the following WHO study in your reference list:    

Maltezou et al. Influenza vaccination policies for health workers in low-income and middle-income countries: A cross-sectional survey, January-March 2020. Vaccine. 2020 Nov 3;38(47):7433-7439

Response: Thank you for sharing the relevant article. We have added this reference in the revised manuscript.

Submission Date

26 January 2023

Reviewer 3 Report

The aim of the study is very interesting, reporting about an important field of public health such as the uptake of influenza vaccine among healthcare workers (HCWs) especially as conducted in this paper. However, an English language revision could be useful for spelling errors, as well as for some the orthographic and formatting oversight.

The abstract is concise and reports the aims and outcomes of the research, although some spelling and phrasing mistakes needs to be revised.

Introduction is interesting, I would suggest to rewrite the last sentence of the first paragraph to make the text more fluent where it refers to the Barchitta et al. study. Furthermore, when discussing vaccination campaigns in the workplace, the authors may want to add a paragraph concerning vaccination uptake outside of hospitals/HCWs, as it would offer a comparison with a different population for their outcome. This could prove instrumental to add to the discussion: the authors suggest vaccination should be mandatory in HCWs, but do not offer a comparison with other categories to explain this.

Materials and methods are well written and have all the information, the study design is thoroughly explained. The Results section is complete and the additional figures and tables make the process clear and understandable. The Discussion and Conclusions offer an insight into the problem and frame the study in the appropriate context (although could be implemented as suggested above).

The bibliography needs to be formatted according to the journal’s style.

Author Response

Dear Reviewer;

Thank you very much for your kind review of our manuscript titled "(vaccines-2183666) "Uptake of influenza vaccine and factors associated with influenza vaccination among healthcare workers in tertiary care hospitals in Bangladesh: A multicenter cross-sectional study." We have addressed your comments and resubmitted the revised version of the manuscript. We (Author & Co. Authors) tried our best to concentrate on the comments/ suggestions. We appreciate your thorough reviews and very useful comments. We gratefully acknowledge the recommendations, that was very helpful in enriching our manuscript.

Please find the point by point comments and address below

The aim of the study is very interesting, reporting about an important field of public health such as the uptake of influenza vaccine among healthcare workers (HCWs) especially as conducted in this paper. However, an English language revision could be useful for spelling errors, as well as for some the orthographic and formatting oversight.

Response: Thank you so much. We are sorry for the spelling mistake and grammatical errors. We carefully reviewed and rechecked the entire manuscript and corrected those.

The abstract is concise and reports the aims and outcomes of the research, although some spelling and phrasing mistakes needs to be revised.

Response: Thank you for pointing the errors, we have corrected spelling mistakes.

Introduction is interesting, I would suggest to rewrite the last sentence of the first paragraph to make the text more fluent where it refers to the Barchitta et al. study.

Response: Thank you for the observation. We have rephrased the sentences and are now better fluent. 

Furthermore, when discussing vaccination campaigns in the workplace, the authors may want to add a paragraph concerning vaccination uptake outside of hospitals/HCWs, as it would offer a comparison with a different population for their outcome. This could prove instrumental to add to the discussion: the authors suggest vaccination should be mandatory in HCWs, but do not offer a comparison with other categories to explain this.

Response: Thank you for the suggestions. We have included a sentence, "The vaccination campaign should not limit to hospital settings for HCWs but also other risky populations such as pregnant women and the aged population," in the revised manuscript. There is no free flu vaccination provision in Bangladesh, so it isn't easy to compare. We can recommend the influenza vacation for the vulnerable population, such as pregnant women and the aged population.

Materials and methods are well written and have all the information, the study design is thoroughly explained. The Results section is complete and the additional figures and tables make the process clear and understandable. The Discussion and Conclusions offer an insight into the problem and frame the study in the appropriate context (although could be implemented as suggested above).

Response: Thank you, we appreciate your feedback,

The bibliography needs to be formatted according to the journal’s style.

Response: We have revised the reference list according to the journal guideline.

Submission Date

26 January 2023

Round 2

Reviewer 1 Report

Title:

“Uptake of influenza vaccine and factors associated with influenza vaccination among healthcare workers in tertiary care hospitals in Bangladesh: A multicenter cross-sectional study”

1.     First revision    January 19, 2023

The manuscript is interesting, and the intent of this revision is to improve this manuscript. Some suggestions may be indicated.

1. In this study, to demonstrate that participants are a sample representative of the healthcare workers of tertiary care hospitals in Bangladesh is important. In order to indicate this aspect, a detail explanation of the sampling design, power, α, confidence level, sample size, etc. could be mentioned in the manuscript. For example, see (1)

2. The participation rate is not described, and this is needed for an evaluation of the study. On the other hand, a comparison between participants and non-participants could understand better the study

3. Considering that the design of the study, confidence limits of several outcomes such as the prevalence influenza vaccination may be reported.

4. The potential confounding factors analysed could be indicated in the material and methods.

5. Sex is not included in the logistic regression model (Table 3). However, this variable may be not a significative but may be a confounding factor.

6. In Table 3, all variables were analysed together. This may be considered the nominated “table 2 fallacy” (2). Independent variables (healthcare workers type, age,…..) could have some different confounding factors, and Table 3 may be misunderstood.

7. There are some limitations of this study which are not mentioned in the manuscript, including the logistic models. The construction of multivariable models could be ameliorated with a Direct Acyclic Graphic (DAGs) approach, which permits a better approach to adjust for confounders factors (3, 4). In addition, an objective measure of influenza vaccination is not presented such as a vaccination register.  

8.   References of the manuscript should follow the journal’s reference guidelines.

References

1. Riccio, M.; Marte, M.; Imeshtari, V.; Vezza, F.; Barletta, V.I.; Shaholli, D.; Colaprico, C.; Di Chiara, M.; Caresta, E.; Terrin, G.; Papoff, P.; La Torre, G. Analysis of Knowledge, Attitudes and Behaviours of Health Care Workers towards Vaccine-Preventable Diseases and Recommended Vaccinations: An Observational Study in a Teaching Hospital. Vaccines 2023, 11, 196. https://doi.org/10.3390/vaccines11010196

2. Westreich D1, Greenland S. The table 2 fallacy: presenting and interpreting confounder and modifier coefficients. Am J Epidemiol. 2013;177:292-298.

3. Greenland S, Pearl J, Robins JM. Causal diagrams for epidemiologic research. Epidemiology. 1999;10:37–48. pmid:9888278

4. Textor J, van der Zander B, Gilthorpe MS, Liskiewicz M, Ellison GT. Robust causal inference using directed acyclic graphs: the R package ’dagitty’. Int J Epidemiol. 2016;45:1887–1894. pmid:28089956.

2. Second revision                                             January 27, 2023

This is the new revision of the manuscript after the response of the authors to the suggestions from the first revision. The authors have addressed several of the suggestions and commends of the first revision. However, the main suggestion about the characteristics of sample design has insufficient response.

 1.With respect to “a detail explanation of the sampling design, power, α, confidence level, sample size, etc.,” the authors only mention the sample size (n=2130). However, no the sampling design, power, α, etc. are indicated, and the response is insufficient to consider that the included sample was representative of the healthcare workers of tertiary care hospitals in Bangladesh. ¿How the sample was calculated?

2. The response rate is indicated.

3. The confidence limits are reported but only for influenza vaccine. The CI could be reported in other results.

4. The potential confounding factors analysed have been reported.

5. Sex is included in the logistic regression model.

6. “Table 2 fallacy” is not addressed by the authors

7. The authors included only a limitation: no objective measure of influenza vaccination was presented.

8.   References of the manuscript are changed but the journal’s reference guidelines are not followed.

Author Response

  1. Second revision January 27, 2023

This is the new revision of the manuscript after the response of the authors to the suggestions from the first revision. The authors have addressed several of the suggestions and commends of the first revision. However, the main suggestion about the characteristics of sample design has insufficient response.

Dear Reviewer;

Thank you very much for your kind review again of our manuscript titled. We have addressed your comments and resubmitted the revised version of the manuscript. We have tried our best to address the comments / suggestions. We appreciate your thorough reviews and very useful comments. We gratefully acknowledge the recommendations, that was very helpful in enriching our manuscript.

  • With respect to “a detail explanation of the sampling design, power, α, confidence level, sample size, etc.,” the authors only mention the sample size (n=2130). However, no the sampling design, power, α, etc. are indicated, and the response is insufficient to consider that the included sample was representative of the healthcare workers of tertiary care hospitals in Bangladesh. ¿How the sample was calculated?

Response: Thank you for your further query. Respondents were selected proportionally from each hospital. We collected the entire hospital staff list first, then randomly enrolled the study particpant for this study. We did not use the sample size calculation formula, and for this, we didn’t mention power, α, etc. We considered 25% of each participant category would be sufficient for representation from each hospital. We have revised the writing on pages 2-3, lines 94-97 as below:

“We enrolled participants proportionally from all respondent groups and hospitals. First, we collected respondent-specific total employee numbers from each hospital and randomly interviewed 25% of each category who were available at the time of data collection and willing to participate.

For example, hospital-1 had 142 physicians, 236 nurses, 51 cleaning staff and 39 administrative staff. So, for that specific hospital, we enrolled 142*25% = 31 physicians, 236*25% = 59 nurses, 51*25% = 13 cleaning staff and 39*25% = 10 administrative staff for the interview.

  • The response rate is indicated.

Response: Thank you, we have added the response rate in the earlier version.

  • The confidence limits are reported but only for influenza vaccine. The CI could be reported in other results.

Response: Thank you for your observation. We have included the CI for ‘Know about influenza vaccination’, ‘Place of free vaccine’, ‘The desire for taking influenza vaccine upon availability (among unvaccinated HCWs)’, ‘Recommended influenza vaccine for HCWs’ and ‘Should compulsory to get the influenza vaccine’ in page 5, lines 163-177.

  • The potential confounding factors analysed have been reported.

Response: Thank you, we have added the response in the earlier version.

  • Sex is included in the logistic regression model.

Response: Thank you, we have added the response in the earlier version.

  • “Table 2 fallacy” is not addressed by the authors

Response: Thank you very much for redirecting the issue. We have tried to clarify it in detail in the 'Statistical Analysis section' on page 3, lines 121-130. We performed the statistical analysis method' multivariate logistic regression model' to determine the association. First, we checked multicollinearity between independent variables for the univariate model to avoid misinterpretation. Then, a p-value of <0.25 in the univariate model was considered for the final model. Finally, we presented the results considering the adjusted odds ratio (AOR) with a 95% confidence interval (CI) and p < 0.05 to demonstrate a statistically significant association.

  • The authors included only a limitation: no objective measure of influenza vaccination was presented.

Response: Thank you, we have also added other limitations in the revised manuscript.

  1. References of the manuscript are changed but the journal’s reference guidelines are not followed.

 Response: Thank you, we have rechecked and revised the reference according to the journal guideline.

Kind regards,

Md. Golam Dostogir Harun

Round 3

Reviewer 1 Report

Title:

“Uptake of influenza vaccine and factors associated with influenza vaccination among healthcare workers in tertiary care hospitals in Bangladesh: A multicenter cross-sectional study”

1.     First revision    January 19, 2023

The manuscript is interesting, and the intent of this revision is to improve this manuscript. Some suggestions may be indicated.

1. In this study, to demonstrate that participants are a sample representative of the healthcare workers of tertiary care hospitals in Bangladesh is important. In order to indicate this aspect, a detail explanation of the sampling design, power, α, confidence level, sample size, etc. could be mentioned in the manuscript. For example, see (1)

2. The participation rate is not described, and this is needed for an evaluation of the study. On the other hand, a comparison between participants and non-participants could understand better the study

3. Considering that the design of the study, confidence limits of several outcomes such as the prevalence influenza vaccination may be reported.

4. The potential confounding factors analysed could be indicated in the material and methods.

5. Sex is not included in the logistic regression model (Table 3). However, this variable may be not a significative but may be a confounding factor.

6. In Table 3, all variables were analysed together. This may be considered the nominated “table 2 fallacy” (2). Independent variables (healthcare workers type, age,…..) could have some different confounding factors, and Table 3 may be misunderstood.

7. There are some limitations of this study which are not mentioned in the manuscript, including the logistic models. The construction of multivariable models could be ameliorated with a Direct Acyclic Graphic (DAGs) approach, which permits a better approach to adjust for confounders factors (3, 4). In addition, an objective measure of influenza vaccination is not presented such as a vaccination register.  

8.   References of the manuscript should follow the journal’s reference guidelines.

References

1. Riccio, M.; Marte, M.; Imeshtari, V.; Vezza, F.; Barletta, V.I.; Shaholli, D.; Colaprico, C.; Di Chiara, M.; Caresta, E.; Terrin, G.; Papoff, P.; La Torre, G. Analysis of Knowledge, Attitudes and Behaviours of Health Care Workers towards Vaccine-Preventable Diseases and Recommended Vaccinations: An Observational Study in a Teaching Hospital. Vaccines 2023, 11, 196. https://doi.org/10.3390/vaccines11010196

2. Westreich D1, Greenland S. The table 2 fallacy: presenting and interpreting confounder and modifier coefficients. Am J Epidemiol. 2013;177:292-298.

3. Greenland S, Pearl J, Robins JM. Causal diagrams for epidemiologic research. Epidemiology. 1999;10:37–48. pmid:9888278

4. Textor J, van der Zander B, Gilthorpe MS, Liskiewicz M, Ellison GT. Robust causal inference using directed acyclic graphs: the R package ’dagitty’. Int J Epidemiol. 2016;45:1887–1894. pmid:28089956.

2. Second revision                                             January 27, 2023

This is the new revision of the manuscript after the response of the authors to the suggestions from the first revision. The authors have addressed several of the suggestions and commends of the first revision. However, the main suggestion about the characteristics of sample design has insufficient response.

 1.With respect to “a detail explanation of the sampling design, power, α, confidence level, sample size, etc.,” the authors only mention the sample size (n=2130). However, no the sampling design, power, α, etc. are indicated, and the response is insufficient to consider that the included sample was representative of the healthcare workers of tertiary care hospitals in Bangladesh. ¿How the sample was calculated?

2. The response rate is indicated.

3. The confidence limits are reported but only for influenza vaccine. The CI could be reported in other results.

4. The potential confounding factors analysed have been reported.

5. Sex is included in the logistic regression model.

6. “Table 2 fallacy” is not addressed by the authors

7. The authors included only a limitation: no objective measure of influenza vaccination was presented.

8.   References of the manuscript are changed but the journal’s reference guidelines are not followed.

Third revision                                                     March 1, 2023

The authors mention that sampling design, power, or α were not estimated and the sample size was obtained from a peculiar way. Considering this situation, the sample may be not representative of the healthcare workers of tertiary hospitals in Bangladesh.

Again, two recommendations may be indicated:

1.     Include a more detailed explanation of the sample size procedures in Material and Methods. This paragraph needs to be included.  

“For example, hospital-1 had 142 physicians, 236 nurses, 51 cleaning staff and 39 administrative staff. So, for that specific hospital, we enrolled 142*25% = 31 physicians, 236*25% = 59 nurses, 51*25% = 13 cleaning staff and 39*25% = 10 administrative staff for the interview.”

2.     Follow the journal’s reference guidelines

Author Response

Thank you very much for your review and remarks. We have addressed two comments accordingly. 

  1. Include a more detailed explanation of the sample size procedures in Material and Methods. This paragraph needs to be included.

Sample size and sampling procedures:

“For example, hospital-1 had 142 physicians, 236 nurses, 51 cleaning staff and 39 administrative staff. So, for that specific hospital, we enrolled 142*25% = 31 physicians, 236*25% = 59 nurses, 51*25% = 13 cleaning staff and 39*25% = 10 administrative staff for the interview.”

Response: Thank you. We have addressed your comments accordingly. We mentioned that we enrolled 25% HCWs proportionately to ensure the representativeness of each category. We have added a sub-paragraph in the material and methods section of the revised manuscript on page 3, lines 101-103. 

We enrolled participants proportionately from all respondent groups and hospitals. We collected respondent-specific total employee numbers from each hospital and randomly interviewed 25% of each respondent category. We enrolled a total of 2046 HCWs for this study that included 526 physicians, 934 nurses, 268 cleaning workers and 318 administrative staff. We considered both permanent and temporary staff who were available at the time of data collection and willing to participate”.

2. Follow the journal’s reference guidelines

Response: Thank you for your remarks. We have updated the reference style according to MDPI guidelines.

Best regards,

Md. Golam Dostogir Harun